# Lower Bounds on the Depth of Integral ReLU Neural Networks via Lattice Polytopes

**Christian Haase**
Freie Universität Berlin, Germany
`haase@math.fu-berlin.de`

**Christoph Hertrich**
London School of Economics and Political Science, UK
`c.hertrich@lse.ac.uk`

**Georg Loho**
University of Twente, Netherlands
`g.loho@utwente.nl`

## Abstract

We prove that the set of functions representable by ReLU neural networks with integer weights strictly increases with the network depth while allowing arbitrary width. More precisely, we show that $\lceil \log_2(n) \rceil$ hidden layers are indeed necessary to compute the maximum of $n$ numbers, matching known upper bounds. Our results are based on the known duality between neural networks and Newton polytopes via tropical geometry. The integrality assumption implies that these Newton polytopes are lattice polytopes. Then, our depth lower bounds follow from a parity argument on the normalized volume of faces of such polytopes.

## 1 Introduction

Classical results in the area of understanding the expressivity of neural networks are so-called *universal approximation theorems* (Cybenko, 1989; Hornik, 1991). They state that shallow neural networks are already capable of approximately representing every continuous function on a bounded domain. However, in order to gain a complete understanding of what is going on in modern neural networks, we would also like to answer the following question: what is the precise set of functions we can compute *exactly* with neural networks of a certain depth? For instance, insights about exact representability have recently boosted our understanding of the computational complexity to train neural networks in terms of both, algorithms (Arora et al., 2018; Khalife & Basu, 2022) and hardness results (Goel et al., 2021; Froese et al., 2022; Bertschinger et al., 2022).

Arguably, the most prominent activation function nowadays is the *rectified linear unit* (ReLU) (Glorot et al., 2011; Goodfellow et al., 2016). While its popularity is primarily fueled by intuition and empirical success, replacing previously used smooth activation functions like sigmoids with ReLUs has some interesting implications from a mathematical perspective: suddenly methods from discrete geometry studying piecewise linear functions and polytopes play a crucial role in understanding neural networks (Arora et al., 2018; Zhang et al., 2018; Hertrich et al., 2021) supplementing the traditionally dominant analytical point of view.

A fundamental result in this direction is by Arora et al. (2018), who show that a function is representable by a ReLU neural network if and only if it is *continuous and piecewise linear* (CPWL). Moreover, their proof implies that $\lceil \log_2(n + 1) \rceil$ many hidden layers are sufficient to represent every CPWL function with $n$-dimensional input. A natural follow-up question is the following: is this logarithmic number of layers actually necessary or can shallower neural networks already represent all CPWL functions?

Hertrich et al. (2021) conjecture that the former alternative is true. More precisely, if $\text{ReLU}_n(k)$ denotes the set of CPWL functions defined on $\mathbb{R}^n$ and computable with $k$ hidden layers, the conjecture can be formulated as follows:

**Conjecture 1** (Hertrich et al. (2021)). $\text{ReLU}_n(k-1) \subsetneq \text{ReLU}_n(k)$ *for all* $k \leq \lceil \log_2(n+1) \rceil$.

Note that $\text{ReLU}_n(\lceil \log_2(n+1) \rceil)$ is the entire set of CPWL functions defined on $\mathbb{R}^n$ by the result of Arora et al. (2018).

While Hertrich et al. (2021) provide some evidence for their conjecture, it remains open for every input dimension $n \geq 4$. Even more drastically, there is not a single CPWL function known for which one can prove that two hidden layers are not sufficient to represent it. Even for a function as simple as $\max\{0, x_1, x_2, x_3, x_4\}$, it is unknown whether two hidden layers are sufficient.

In fact, $\max\{0, x_1, x_2, x_3, x_4\}$ is not just an arbitrary example. Based on a result by Wang & Sun (2005), Hertrich et al. (2021) show that their conjecture is equivalent to the following statement.

**Conjecture 2** (Hertrich et al. (2021)). *For* $n = 2^k$*, the function* $\max\{0, x_1, \ldots, x_n\}$ *is not contained in* $\text{ReLU}_n(k)$.

This reformulation gives rise to interesting interpretations in terms of two elements commonly used in practical neural network architectures: *max-pooling* and *maxout*. Max-pooling units are used between (ReLU or other) layers and simply output the maximum of several inputs (that is, "pool" them together). They do not contain trainable parameters themselves. In contrast, maxout networks are an alternative to (and in fact a generalization of) ReLU networks. Each neuron in a maxout network outputs the maximum of several (trainable) affine combinations of the outputs in the previous layer, in contrast to comparing a single affine combination with zero as in the ReLU case.

Thus, the conjecture would imply that one needs in fact logarithmically many ReLU layers to replace a max-pooling unit or a maxout layer, being a theoretical justification that these elements are indeed more powerful than pure ReLU networks.

## 1.1 OUR RESULTS

In this paper we prove that the conjecture by Hertrich et al. (2021) is true for all $n \in \mathbb{N}$ under the additional assumption that all weights in the neural network are restricted to be integral. In other words, if $\text{ReLU}_n^{\mathbb{Z}}(k)$ is the set of functions defined on $\mathbb{R}^n$ representable with $k$ hidden layers and only integer weights, we show the following.

**Theorem 3.** *For* $n = 2^k$*, the function* $\max\{0, x_1, \ldots, x_n\}$ *is not contained in* $\text{ReLU}_n^{\mathbb{Z}}(k)$.

Proving Theorem 3 is our main contribution. The overall strategy is highlighted in Section 1.2. We put all puzzle pieces together and provide a formal proof in Section 4.

The arguments in Hertrich et al. (2021) can be adapted to show that the equivalence between the two conjectures is also valid in the integer case. Thus, we obtain that adding more layers to an integral neural network indeed increases the set of representable functions up to a logarithmic number of layers. A formal proof can be found in Section 4.

**Corollary 4.** $\text{ReLU}_n^{\mathbb{Z}}(k-1) \subsetneq \text{ReLU}_n^{\mathbb{Z}}(k)$ *for all* $k \leq \lceil \log_2(n+1) \rceil$.

To the best of our knowledge, our result is the first non-constant (namely logarithmic) lower bound on the depth of ReLU neural networks without any restriction on the width. Without the integrality assumption, the best known lower bound remains two hidden layers Mukherjee & Basu (2017), which is already valid for the simple function $\max\{0, x_1, x_2\}$.

While the integrality assumption is rather implausible for practical neural network applications where weights are usually tuned by gradient descent, from a perspective of analyzing the theoretical expressivity, the assumption is arguably plausible. To see this, suppose a ReLU network represents the function $\max\{0, x_1, \ldots, x_n\}$. Then every fractional weight must either cancel out or add up to some integers with other fractional weights, because every linear piece in the final function has only integer coefficients. Hence, it makes sense to assume that no fractional weights exist in the first place. However, unfortunately, this intuition cannot easily be turned into a proof because it might happen that combinations of fractional weights yield integer coefficients which could not be achieved without fractional weights.

## 1.2 OUTLINE OF THE ARGUMENT

The first ingredient of our proof is to apply previous work about connections between neural networks and tropical geometry, initiated by Zhang et al. (2018). Every CPWL function can be decomposed as a difference of two convex CPWL functions. Convex CPWL functions admit a neat duality to certain polytopes, known as *Newton polytopes* in tropical geometry. Given any neural network, this duality makes it possible to construct a pair of Newton polytopes which uniquely determines the CPWL function represented by the neural network. These Newton polytopes are constructed, layer by layer, from points by alternatingly taking Minkowski sums and convex hulls. The number of alternations corresponds to the depth of the neural network. Thus, analyzing the set of functions representable by neural networks with a certain depth is equivalent to understanding which polytopes can be constructed in this manner with a certain number of alternations (Theorem 8).

Having translated the problem into the world of polytopes, the second ingredient is to gain a better understanding of the two operations involved in the construction of these polytopes: Minkowski sums and convex hulls. We show for each of the two operations that the result can be subdivided into polytopes of easier structure: For the Minkowski sum of two polytopes, each cell in the subdivision is an *affine product* of faces of the original polytopes, that is, a polytope which is affinely equivalent to a Cartesian product (Proposition 9). For the convex hull of two polytopes, each cell is a *join* of two faces of the original polytopes, that is, a convex hull of two faces whose affine hulls are *skew* to each other, which means that they do not intersect and do not have parallel directions (Proposition 10).

Finally, with these subdivisions at hand, our third ingredient is the volume of these polytopes. Thanks to our integrality assumption, all coordinates of all vertices of the relevant polytopes are integral, that is, these polytopes are *lattice polytopes*. For lattice polytopes, one can define the so-called *normalized volume* (Section 2.3). This is a scaled version of the Euclidean volume with scaling factor depending on the affine hull of the polytope. It is constructed in such a way that it is integral for all lattice polytopes. Using the previously obtained subdivisions, we show that the normalized volume of a face of dimension at least $2^k$ of a polytope corresponding to a neural network with at most $k$ hidden layers has always even normalized volume (Theorem 16). This implies that not all lattice polytopes can be constructed this way. Using again the tropical geometry inspired duality between polytopes and functions (Theorem 8), we translate this result back to the world of CPWL functions computed by neural networks and obtain that $k$ hidden layers are not sufficient to compute $\max\{0, x_1, \ldots, x_{2^k}\}$, proving Theorem 3.

## 1.3 BEYOND INTEGRAL WEIGHTS

Given the integrality assumption, a natural thought is whether one can simply generalize our results to rational weights by multiplying all weights of the neural network with the common denominator. Unfortunately this does not work. Our proof excludes that an integral ReLU network with $k$ hidden layers can compute $\max\{0, x_1, \ldots, x_{2^k}\}$, but it does not exclude the possibility to compute $2 \cdot \max\{0, x_1, \ldots, x_{2^k}\}$ with integral weights. In particular, dividing the weights of the output layer by two might result in a half-integral neural network computing $\max\{0, x_1, \ldots, x_{2^k}\}$.

In order to tackle the conjecture in full generality, that is, for arbitrary weights, arguing via the parity of volumes seems not to be sufficient. The parity argument is inherently discrete and suffers from the previously described issue. Nevertheless, we are convinced that the techniques of this paper do in fact pave the way towards resolving the conjecture in full generality. In particular, the subdivisions constructed in Section 3 are valid for arbitrary polytopes and not only for lattice polytopes. Hence, it is conceivable that one can replace the parity of normalized volumes with a different invariant which, applied to the subdivisions, yields a general depth-separation for the non-integer case.

## 1.4 FURTHER RELATIONS TO PREVIOUS WORK

Similar to our integrality assumption, also Hertrich et al. (2021) use an additional assumption and prove their conjecture in a special case for so-called $H$-*conforming* neural networks. Let us note that the two assumptions are incomparable: there are $H$-conforming networks with non-integral weights as well as integral neural networks which are not $H$-conforming.

Our results are related to (but conceptually different from) so-called *trade-off* results between depth and width, showing that a slight decrease of depth can exponentially increase the required width to

maintain the same (exact or approximate) expressive power. Telgarsky (2015; 2016) proved the first results of this type and Eldan & Shamir (2016) even proved an exponential separation between two and three layers. Lots of other improvements and related results have been established (Arora et al., 2018; Daniely, 2017; Hanin, 2019; Hanin & Sellke, 2017; Liang & Srikant, 2017; Nguyen et al., 2018; Raghu et al., 2017; Safran & Shamir, 2017; Safran et al., 2019; Vardi et al., 2021; Yarotsky, 2017). In contrast, we focus on exact representations, where we show an even more drastic trade-off: decreasing the depth from logarithmic to sublogarithmic yields that no finite width is sufficient at all any more.

The duality between CPWL functions computed by neural networks and Newton polytopes inspired by tropical geometry has been used in several other works about neural networks before (Maragos et al., 2021), for example to analyze the shape of decision boundaries (Alfarra et al., 2020; Zhang et al., 2018) or to count and bound the number of linear pieces (Charisopoulos & Maragos, 2018; Hertrich et al., 2021; Montúfar et al., 2022).

## 2 PRELIMINARIES

We write $[n] \coloneqq \{1, 2, \ldots, n\}$ for the set of natural numbers up to $n$ (without zero).

### 2.1 ReLU NEURAL NETWORKS

For any $n \in \mathbb{N}$, let $\sigma \colon \mathbb{R}^n \to \mathbb{R}^n$ be the component-wise *rectifier* function

$$\sigma(x) = (\max\{0, x_1\}, \max\{0, x_2\}, \ldots, \max\{0, x_n\}).$$

For any *number of hidden layers* $k \in \mathbb{N}$, a $(k+1)$-*layer feedforward neural network with rectified linear units* (ReLU neural network) is given by $k + 1$ affine transformations $T^{(\ell)} \colon \mathbb{R}^{n_{\ell-1}} \to \mathbb{R}^{n_\ell}$, $x \mapsto A^{(\ell)} x + b^{(\ell)}$, for $\ell \in [k+1]$. It is said to *compute* or *represent* the function $f \colon \mathbb{R}^{n_0} \to \mathbb{R}^{n_{k+1}}$ given by

$$f = T^{(k+1)} \circ \sigma \circ T^{(k)} \circ \sigma \circ \cdots \circ T^{(2)} \circ \sigma \circ T^{(1)}.$$

The matrices $A^{(\ell)} \in \mathbb{R}^{n_\ell \times n_{\ell-1}}$ are called the *weights* and the vectors $b^{(\ell)} \in \mathbb{R}^{n_\ell}$ are the *biases* of the $\ell$-th layer. The number $n_\ell \in \mathbb{N}$ is called the *width* of the $\ell$-th layer. The maximum width of all hidden layers $\max_{\ell \in [k]} n_\ell$ is called the *width* of the neural network. Further, we say that the neural network has *depth* $k + 1$ and *size* $\sum_{\ell=1}^{k} n_\ell$.

Often, neural networks are represented as layered, directed, acyclic graphs where each dimension of each layer (including *input layer* $\ell = 0$ and *output layer* $\ell = k + 1$) is one vertex, weights are arc labels, and biases are node labels. The vertices of this graph are called *neurons*. Each neuron computes an affine transformation of the outputs of their predecessors, applies the ReLU function on top of that, and outputs the result.

### 2.2 POLYTOPES & LATTICES

We give a brief overview of necessary notions for polytopes and lattices and refer to (Schrijver, 1986, Ch. 4,7-9) for further reading. For two arbitrary sets $X, Y \subseteq \mathbb{R}^n$, one can define their *Minkowski sum* $X + Y = \{x + y \mid x \in X, y \in Y\}$. By $\mathrm{Span}(X)$ we denote the usual linear hull of the set $X$, that is, the smallest linear subspace containing $X$. The affine hull $\mathrm{Aff}(X)$ is the smallest affine subspace containing $X$. Finally, with a set $X$ we associate a linear space $\mathrm{Lin}(X) = \mathrm{Span}\{x - y \mid x, y \in X\}$, which is $\mathrm{Aff}(X)$ shifted to the origin. Note that $\mathrm{Lin}(X)$ is usually a strict subspace of $\mathrm{Span}(X)$, unless $\mathrm{Aff}(X)$ contains the origin, in which case all three notions coincide.

For a set $X \subset \mathbb{R}^n$, its *convex hull* is

$$\mathrm{conv}(X) = \left\{ \sum_{s \in S} \lambda_s s \mid S \subseteq X \text{ finite}, \lambda_s \geq 0 \text{ for all } s \in S, \sum_{s \in S} \lambda_s = 1 \right\}.$$

A *polytope* is the convex hull of a finite set $V \subset \mathbb{R}^n$. Given a polytope $P \subset \mathbb{R}^n$, a *face* of $P$ is a set of the form $\arg\min\{c^\top x \mid x \in P\}$ for some $c \in \mathbb{R}^n$; a face of a polytope is again a polytope. The *dimension* $\dim(P)$ of $P$ is the dimension of $\mathrm{Lin}(P)$. By convention, we also call the empty set a

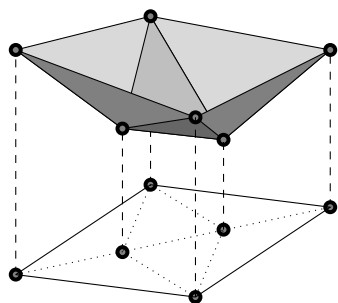

Figure 1: A regular subdivision: the quadrilateral in $\mathbb{R}^2$ is subdivided into six full-dimensional cells which arise as projections of lower faces of the convex hull of the lifted points in $\mathbb{R}^3$.

face of $P$ of dimension $-1$. A face of dimension $0$ is a *vertex* and a face of dimension $\dim(P) - 1$ is a *facet*. If all vertices belong to $\mathbb{Q}^n$, we call $P$ *rational*; even more, $P$ is a *lattice polytope* if all its vertices are *integral*, that means they lie in $\mathbb{Z}^n$.

An important example of a (lattice) polytope is the *simplex* $\Delta_0^n = \mathrm{conv}\{0, e_1, e_2, \ldots, e_n\} \subseteq \mathbb{R}^n$ spanned by the origin and all $n$ standard basis vectors.

Finally, we define *(regular polyhedral) subdivisions* of a polytope. Let $\tilde{P} \subset \mathbb{R}^{n+1}$ be a polytope and let $P \subset \mathbb{R}^n$ be its image under the projection $\mathrm{proj}_{[n]} \colon \mathbb{R}^{n+1} \to \mathbb{R}^n$ forgetting the last coordinate. A *lower face* of $\tilde{P}$ is a set of minimizers with respect to a linear objective $(c^\top, 1) \in \mathbb{R}^n \times \mathbb{R}$. The projection of the lower faces of $\tilde{P}$ to $P \subset \mathbb{R}^n$ forms a *subdivision* of $P$, that is, a collection of polytopes which intersect in common faces and cover $P$. An example of a subdivision of a polytope in the plane is given in Figure 1.

A *lattice* is a subset of $\mathbb{R}^n$ of the form

$$L = \{B \cdot z \mid z \in \mathbb{Z}^p\}$$

for some matrix $B \in \mathbb{R}^{n \times p}$ with linearly independent columns. In this case, we call the columns $b^{(1)}, \ldots, b^{(p)}$ of $B$ a *lattice basis*. Every element in $L$ can be written uniquely as a linear combination of $b^{(1)}, \ldots, b^{(p)}$ with integer coefficients. A choice of lattice basis identifies $L$ with $\mathbb{Z}^p$.

The classic example of a lattice is $\mathbb{Z}^n$ itself. In this paper, we will only work with lattices that arise as

$$\{x \in \mathbb{Z}^n \mid A \cdot x = 0\} \ ,$$

for some $A \in \mathbb{Q}^{q \times n}$, that is, the intersection of $\mathbb{Z}^n$ with a rational subspace of $\mathbb{R}^n$.

For example, the vectors $\binom{2}{1}, \binom{1}{2}$ form a basis of $\mathbb{R}^2$ as a vector space, but they do not form a lattice basis of $\mathbb{Z}^2$. Instead, they generate the smaller lattice $\{x \in \mathbb{Z}^2 \mid 3 \text{ divides } x_1 + x_2\}$. Choosing two bases for the same lattice yields a change of bases matrix with integral entries whose inverse is also integral. It must therefore have determinant $\pm 1$.

We will mainly deal with *affine lattices*, that is, sets of the form $K = L + v$ where $L \subset \mathbb{R}^n$ is a lattice and $v \in \mathbb{R}^n$. Then, $b_0, \ldots, b_r \in K$ form an *affine lattice basis* if $b_1 - b_0, \ldots, b_r - b_0$ is a (linear) lattice basis of the linear lattice $L = K - K$ parallel to $L$, that is, every element of $K$ is a unique affine combination of $b_0, \ldots, b_r$ with integral coefficients.

### 2.3 Normalized volume

The main tool in our proof is the *normalized volume* of faces of (Newton) polytopes. We measure the volume of a (possibly lower dimensional) rational polytope $P \subset \mathbb{R}^n$ inside its affine hull $\mathrm{Aff}(P)$. For this, we choose a lattice basis of $\mathrm{Lin}(P) \cap \mathbb{Z}^n$ inside the linear space $\mathrm{Lin}(P)$ parallel to $\mathrm{Aff}(P)$. Since $P$ is a rational polytope, this lattice basis gives rise to a linear transform from $\mathrm{Lin}(P)$ onto $\mathbb{R}^r$ ($r = \dim P$) by mapping the lattice basis vectors to the standard unit vectors. Now, the *normalized volume* $\mathrm{Vol}(P)$ of $P$ is $r!$ times the Euclidean volume of its image under this transformation. The scaling factor $r!$ is chosen so that, for each $n \geq 1$, the normalized volume of the simplex $\Delta_0^n$ is one.

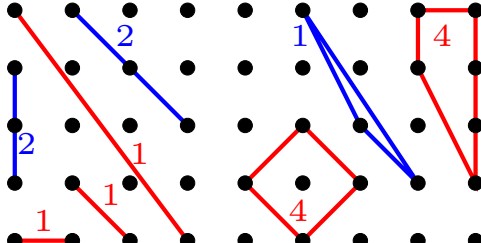 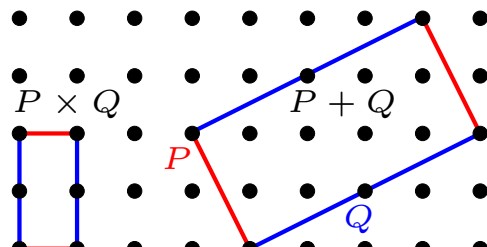

Figure 2: Some lattice segments and lattice polygons with their normalized volumes. Note that the normalized volume of a lattice polytope differs from its Euclidean volume as it is measured with respect to the induced lattice in its affine hull.

Figure 3: The normalized volume of an affine product of $P$ and $Q$ can be calculated as the volume of the Cartesian product times the integral absolute value of the determinant of the linear map sending $P \times Q$ (left) to $P + Q$ (right): $\mathrm{Vol}(P + Q) = \binom{1+1}{1} \cdot \left|\det\left(\begin{smallmatrix} -1 & 2 \\ 2 & 1 \end{smallmatrix}\right)\right| \cdot \mathrm{Vol}(P) \cdot \mathrm{Vol}(Q)$.

The fact that any two lattice bases differ by an integral matrix with integral inverse (and hence determinant $\pm 1$) ensures that this is well defined. For full-dimensional polytopes, this yields just a scaled version of the Euclidean volume. But for lower-dimensional polytopes, our normalization with respect to the lattice differs from the Euclidean normalization (cf. Figure 2). For us, the crucial property is that every lattice polytope has integral volume (see, e.g., (Beck & Robins, 2007, §3.5,§5.4)). This will allow us to argue using divisibility properties of volumes.

We give visualizations of the following fundamental statements and defer the actual proofs to the appendix.

**Lemma 5.** *Let $P, Q \subset \mathbb{R}^n$ be two lattice polytopes with $i = \dim(P)$ and $j = \dim(Q)$. If we have $\dim(P + Q) = \dim(P) + \dim(Q)$ then*

$$\binom{i + j}{i} \cdot \mathrm{Vol}(P) \cdot \mathrm{Vol}(Q) \ \text{ divides } \ \mathrm{Vol}(P + Q) \, .$$

If $P$ and $Q$ fulfill the assumptions of Lemma 5, then we call $P + Q$ an *affine product* of $P$ and $Q$. This is equivalent to the definition given in the introduction; we visualize the Cartesian product $P \times Q$ for comparison in Figure 3.

**Lemma 6.** *Let $P, Q \subset \mathbb{R}^n$ be two lattice polytopes with $i = \dim(P)$ and $j = \dim(Q)$. If $\dim(\mathrm{conv}(P \cup Q)) = \dim(P) + \dim(Q) + 1$ then*

$$\mathrm{Vol}(P) \cdot \mathrm{Vol}(Q) \ \text{ divides } \ \mathrm{Vol}(\mathrm{conv}(P \cup Q)) \, .$$

If $P$ and $Q$ fulfill the assumptions of Lemma 6, then we call $\mathrm{conv}(P \cup Q)$ the *join* of $P$ and $Q$, compare Figure 4. This definition is equivalent with the one given in the introduction.

## 2.4 NEWTON POLYTOPES AND NEURAL NETWORKS

The first ingredient to prove our main result is to use a previously discovered duality between CPWL functions and polytopes inspired by tropical geometry in order to translate the problem into the world of polytopes.

As a first step, let us observe that we may restrict ourselves to neural networks without biases. For this, we say a function $g \colon \mathbb{R}^n \to \mathbb{R}^m$ is *positively homogeneous* if it satisfies $g(\lambda x) = \lambda g(x)$ for all $\lambda \geq 0$.

**Lemma 7** (Hertrich et al. (2021)). *If a neural network represents a positively homogeneous function, then the same neural network with all bias parameters $b^{(\ell)}$ set to zero represents exactly the same function.*

Since $\max\{0, x_1, \ldots, x_{2^k}\}$ is positively homogeneous, this implies that we can focus on neural networks without biases in order to prove Theorem 3. Functions computed by such neural networks are always positively homogeneous.

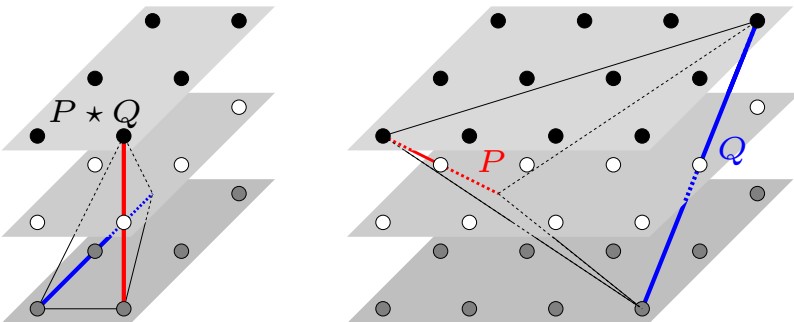

Figure 4: The normalized volume of a join of $P$ and $Q$ can be calculated as the product of the normalized volumes of $P$ and $Q$ times the integral absolute value of the determinant of the linear map sending $P \star Q$ (left) to $\mathrm{conv}(P \cup Q)$ (right): $\mathrm{Vol}(\mathrm{conv}(P \cup Q)) = \left| \det \begin{pmatrix} -2 & 0 & 0 \\ 2 & 1 & -1 \\ 0 & 1 & 1 \end{pmatrix} \right| \cdot \mathrm{Vol}(P) \cdot \mathrm{Vol}(Q)$, where $P \star Q$ denotes an embedding of $P$ and $Q$ in orthogonal subspaces with distance 1.

Let $f$ be a positively homogeneous, convex CPWL function. Convexity implies that we can write $f$ as the maximum of its linear pieces, that is, $f(x) = \max\{a_1^\top x, \ldots, a_p^\top x\}$. The *Newton polytope* $P_f$ corresponding to $f$ is defined as the convex hull of the coefficient vectors, that is, $P_f = \mathrm{conv}\{a_1, \ldots, a_p\}$. It turns out that the two basic operations $+$ and $\max$ (applied pointwise) on functions translate nicely to Minkowski sum and convex hull for the corresponding Newton polytopes: $P_{f+g} = P_f + P_g$ and $P_{\max\{f,g\}} = \mathrm{conv}\{P_f \cup P_g\}$, compare Zhang et al. (2018). Since a neural network is basically an alternation of affine transformations (that is, weighted sums) and maxima computations, this motivates the following definition of classes $\mathcal{P}_k$ of polytopes which intuitively correspond to integral neural networks with $k$ hidden layers. Note that the class $\mathcal{P}_k$ contains polytopes of different dimensions.

We begin with $\mathcal{P}_0$, the set of lattice points. For $k \geq 0$, we define

$$
\begin{aligned}
\mathcal{P}'_{k+1} &= \{\mathrm{conv}(Q_1 \cup Q_2) \mid Q_1, Q_2 \in \mathcal{P}_k\} \ , \\
\mathcal{P}_{k+1} &= \{Q_1 + \cdots + Q_\ell \mid Q_1, \ldots, Q_\ell \in \mathcal{P}'_{k+1}\} \ .
\end{aligned}
\tag{1}
$$

The correspondence of $\mathcal{P}_k$ to a neural network with $k$ hidden layers can be made formal by the following. The difference in the theorem accounts for the fact that $f$ is not necessarily convex, and even if it is, intermediate functions computed by the neural network might be non-convex.

**Theorem 8.** *A positively homogeneous CPWL function $f$ can be represented by an integral $k$-hidden-layer neural network if and only if $f = g - h$ for two convex, positively homogeneous CPWL functions $g$ and $h$ with $P_g, P_h \in \mathcal{P}_k$.*

A proof of the same theorem for the non-integral case can be found in Hertrich (2022, Thm. 3.35). A careful inspection of the proof therein reveals that it carries over to the integral case. For the sake of completeness, we provide a proof in the appendix.

## 3 Subdividing Minkowski Sums and Convex Hulls

The purpose of this section is to develop the main geometric tool of our proof: subdividing Minkowski sums and unions into affine products and joins, respectively. More precisely, we show the following two statements. They are valid for *general* polytopes and not only for lattice polytopes.

**Proposition 9.** *For two polytopes $P$ and $Q$ in $\mathbb{R}^n$, there exists a subdivision of $P + Q$ such that each full-dimensional cell is an affine product of a face of $P$ and a face of $Q$.*

**Proposition 10.** *For two polytopes $P$ and $Q$ in $\mathbb{R}^n$, there exists a subdivision of $\mathrm{conv}\{P \cup Q\}$ such that each full-dimensional cell is a join of a face of $P$ and a face of $Q$.*

The strategy to prove these statements is to lift the polytopes $P$ and $Q$ by one dimension to $\mathbb{R}^{n+1}$ in a *generic* way, perform the respective operation (Minkowski sum or convex hull) in this space, and obtain the subdivision by projecting the *lower faces* of the resulting polytope in $\mathbb{R}^{n+1}$ back to $\mathbb{R}^n$.

More precisely, given an $\alpha \in \mathbb{R}^n$ and $\beta \in \mathbb{R}$, we define

$$P^{\mathbf{0}} := \{(p, 0) \mid p \in P\} \quad \text{and} \quad Q^{\alpha, \beta} := \left\{(q, \alpha^\top q + \beta) \mid q \in Q\right\} .$$

Note that the projections $\text{proj}_{[n]}(P^{\mathbf{0}})$ and $\text{proj}_{[n]}(Q^{\alpha, \beta})$ onto the first $n$ coordinates result in $P$ and $Q$, respectively. Moreover, we obtain subdivisions of $P + Q$ and $\text{conv}\{P \cup Q\}$ by projecting down the lower faces of $P^{\mathbf{0}} + Q^{\alpha, \beta}$ and $\text{conv}\{P^{\mathbf{0}} \cup Q^{\alpha, \beta}\}$, respectively, to the first $n$ coordinates. It remains to show that the parameters can be chosen so that the subdivisions have the desired properties.

To this end, we introduce the following notation for faces of the respective polytopes. For each $c \in \mathbb{R}^n \setminus \{0\}$, let

$$F_c^{\mathbf{0}} = \arg\min\left\{(c^\top, 1)z \mid z \in P^{\mathbf{0}}\right\} \quad \text{and} \quad G_c^{\alpha, \beta} = \arg\min\left\{(c^\top, 1)z \mid z \in Q^{\alpha, \beta}\right\} ,$$

as well as

$$F_c = \arg\min\{c^\top x \mid x \in P\} \quad \text{and} \quad G_c = \arg\min\{(c + \alpha)^\top x \mid x \in Q\} .$$

Observe that $F_c = \text{proj}_{[n]}(F_c^{\mathbf{0}})$ and $G_c = \text{proj}_{[n]}(G_c^{\alpha, \beta})$.

With this, we can finally specify what we mean by "choosing $\alpha$ and $\beta$ *generically*": it means that the linear and affine hulls of $F_c$ and $G_c$ intersect as little as possible.

**Lemma 11.** *One can choose $\alpha$ and $\beta$ such that* $\text{Lin}\, F_c \cap \text{Lin}\, G_c = \{\mathbf{0}\}$ *and* $\text{Aff}\, F_c^{\mathbf{0}} \cap \text{Aff}\, G_c^{\alpha, \beta} = \emptyset$ *for every* $c \in \mathbb{R}^n \setminus \{0\}$.

Using the lemma about generic choices of $\alpha$ and $\beta$, we have the tool to prove the existence of the desired subdivisions. The actual proofs are given in the appendix.

## 4  USING NORMALIZED VOLUME TO PROVE DEPTH LOWER BOUNDS

In this section we complete our proof by applying a parity argument on the normalized volume of cells in the subdivisions constructed in the previous section.

Let $\mathcal{Q}_k$ be the set of lattice polytopes $P$ with the following property: for every face $F$ of $P$ with $\dim(F) \geq 2^k$ we have that $\text{Vol}(F)$ is even. Note that the class $\mathcal{Q}_k$ contains polytopes of different dimensions and, in particular, all lattice polytopes of dimension smaller than $2^k$.

Our plan to prove Theorem 3 is as follows. We first show that the classes $\mathcal{Q}_k$ are closed under Minkowski addition and that taking unions of convex hulls of polytopes in $\mathcal{Q}_k$ always gives a polytope in $\mathcal{Q}_{k+1}$. Using this, induction guarantees that $\mathcal{P}_k \subseteq \mathcal{Q}_k$. We then show that adding the simplex $\Delta_0^{2^k}$ to a polytope in $\mathcal{Q}_k$ gives a polytope which is never contained in $\mathcal{Q}_k$. Combining this separation result with $\mathcal{P}_k \subseteq \mathcal{Q}_k$ and Theorem 8 allows us to prove Theorem 3. The general intuition behind most of the proofs in this section is to use the subdivisions constructed in the previous section and argue about the volume of each cell in the subdivision separately, using the lemmas of Section 2.3.

### 4.1  CLOSEDNESS OF $\mathcal{Q}$

**Proposition 12.** *For $P, Q \in \mathcal{Q}_k$, we have that $P + Q \in \mathcal{Q}_k$.*

The proof of Proposition 12 is based on the following lemma.

**Lemma 13.** *If $d := \dim(P + Q) \geq 2^k$, then $\text{Vol}(P + Q)$ is even.*

*Proof of Lemma 13.* By Proposition 9, it follows that $P + Q$ can be subdivided such that each full-dimensional cell $C$ in the subdivision is an affine product of two faces $F$ and $G$ of $P$ and $Q$, respectively. Let $i := \dim(F)$ and $j := \dim(G)$, then $d = i + j$. By Lemma 5, it follows that $\text{Vol}(C) = z \cdot \binom{d}{i} \cdot \text{Vol}(F) \cdot \text{Vol}(G)$ for some $z \in \mathbb{Z}$. We argue now that this quantity is always even. If either $i$ or $j$ is at least $2^k$, then $\text{Vol}(F)$ or $\text{Vol}(G)$ is even, respectively, because $P$ and $Q$ are contained in $\mathcal{Q}_k$. Otherwise, $d = i + j \geq 2^k$, but $i, j < 2^k$. In this case, $\binom{d}{i}$ is always even, which is a direct consequence of Lucas' theorem (Lucas, 1878). In any case, for every cell $C$ in the subdivision, $\text{Vol}(C)$ is even. Therefore, also the total volume of $P + Q$ is even. $\qquad\square$

*Proof of Proposition 12.* To prove the proposition, we need to ensure that not only $P + Q$, but every face of $P + Q$ with dimension at least $2^k$ has even normalized volume. If $F$ is such a face, then, by basic properties of the Minkowski sum, it is the Minkowski sum of two faces $P'$ and $Q'$ of $P$ and $Q$, respectively. Since $\mathcal{Q}_k$ is closed under taking faces, it follows that $P', Q' \in \mathcal{Q}_k$. Hence, applying Lemma 13 to $P'$ and $Q'$, it follows that $F$ has even volume. Doing so for all faces $F$ with dimension at least $2^k$ completes the proof. □

**Proposition 14.** *For $P, Q \in \mathcal{Q}_k$, we have that $\mathrm{conv}(P \cup Q) \in \mathcal{Q}_{k+1}$.*

Again, the heart of the proof lies in the following lemma.

**Lemma 15.** *If $d := \dim(\mathrm{conv}(P \cup Q)) \geq 2^{k+1}$, then $\mathrm{Vol}(\mathrm{conv}(P \cup Q))$ is even.*

We defer the proofs of Lemma 15 and Proposition 14 and to the Appendix as they are analogous to those of Proposition 12 and Lemma 13 building on the respective claims for convex hulls (Lemma 6 and Proposition 10) instead of Minkowski sums.

**Theorem 16.** *For all $k \in \mathbb{N}$ it is true that $\mathcal{P}_k \subseteq \mathcal{Q}_k$.*

*Proof.* We prove this statement by induction on $k$. The class $\mathcal{P}_0$ contains only points, so no polytope in $\mathcal{P}_0$ has a face of dimension at least $2^0 = 1$. Therefore, it trivially follows that $\mathcal{P}_0 \subseteq \mathcal{Q}_0$, settling the induction start. The induction step follows by applying Proposition 12 and Proposition 14 to the definition (1) of the classes $\mathcal{P}_k$. □

## 4.2 THE ODD ONE OUT

The final ingredient for Theorem 3 is to show how $\Delta_0^n$ breaks the structure of $\mathcal{Q}_k$.

**Proposition 17.** *If $P \in \mathcal{Q}_k$ is a polytope in $\mathbb{R}^n$ with $n = 2^k$, then $P + \Delta_0^n \notin \mathcal{Q}_k$.*

*Proof.* Applying Proposition 9 to $P$ and $Q = \Delta_0^n$, we obtain that $P + \Delta_0^n$ can be subdivided such that each full-dimensional cell $C$ is an affine product of a face $F$ of $P$ and a face $G$ of $\Delta_0^n$.

As in the proof of Lemma 13, it follows that all these cells have even volume, with one exception: if $\dim F = 0$ and $\dim G = 2^k$. Revisiting the proof of Proposition 9 shows that there exists exactly one such cell $C$ in the subdivision. This cell is a translated version of $\Delta_0^n$, so it has volume $\mathrm{Vol}(C) = 1$.

Since all cells in the subdivision have even volume except for one cell with odd volume, we obtain that $\mathrm{Vol}(P + \Delta_0^n)$ is odd. Hence, $P + \Delta_0^n$ cannot be contained in $\mathcal{Q}_k$. □

**Theorem 3.** *For $n = 2^k$, the function $\max\{0, x_1, \ldots, x_n\}$ is not contained in $\mathrm{ReLU}_n^{\mathbb{Z}}(k)$.*

*Proof.* Suppose for the sake of a contradiction that there is a neural network with integer weights and $k$ hidden layers computing $f(x) = \max\{0, x_1, \ldots, x_n\}$. Since the Newton polytope of $f$ is $P_f = \Delta_0^n$, Theorem 8 yields the existence of two polytopes $P, Q \in \mathcal{P}_k$ in $\mathbb{R}^n$ with $P + \Delta_0^n = Q$. By Theorem 16 we obtain $P, Q \in \mathcal{Q}_k$. This is a contradiction to Proposition 17. □

From this we conclude that the set of functions representable with integral ReLU neural networks strictly increases when adding more layers.

**Corollary 4.** $\mathrm{ReLU}_n^{\mathbb{Z}}(k-1) \subsetneq \mathrm{ReLU}_n^{\mathbb{Z}}(k)$ *for all $k \leq \lceil \log_2(n+1) \rceil$.*

*Proof.* Note that $k \leq \lceil \log_2(n+1) \rceil$ implies $2^{k-1} \leq n$. Let $f : \mathbb{R}^n \to \mathbb{R}$ be the function $f(x) = \max\{0, x_1, \ldots, x_{2^{k-1}}\}$. By Theorem 3 we get $f \notin \mathrm{ReLU}_n^{\mathbb{Z}}(k-1)$. In contrast, it is not difficult to see that $k - 1$ hidden layers with integer weights are sufficient to compute $\max\{x_1, \ldots, x_{2^{k-1}}\}$, compare for example Zhang et al. (2018); Hertrich (2022). Appending a single ReLU neuron to the output of this network implies $f \in \mathrm{ReLU}_n^{\mathbb{Z}}(k)$. □

ACKNOWLEDGMENTS

Christian Haase has been supported by the German Science Foundation (DFG) under grant HA 4383/8-1.

Christoph Hertrich is supported by the European Research Council (ERC) under the European Union's Horizon 2020 research and innovation programme (grant agreement ScaleOpt–757481).

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

# A  ADDITIONAL PROOFS

**Lemma 5.** *Let $P, Q \subset \mathbb{R}^n$ be two lattice polytopes with $i = \dim(P)$ and $j = \dim(Q)$. If we have $\dim(P + Q) = \dim(P) + \dim(Q)$ then*

$$\binom{i + j}{i} \cdot \mathrm{Vol}(P) \cdot \mathrm{Vol}(Q) \ \ divides \ \ \mathrm{Vol}(P + Q) \,.$$

*Proof.* The assumption $\dim(P + Q) = \dim(P) + \dim(Q)$ is equivalent to $\mathrm{Lin}(P) \cap \mathrm{Lin}(Q) = \{0\}$ so that there is an affine bijection $f \colon \mathrm{Aff}(P) \times \mathrm{Aff}(Q) \to \mathrm{Aff}(P + Q)$ mapping lattice points to lattice points. Thus, the volume of $P + Q$ equals the volume of $P \times Q$ times the (integral) determinant of $f$ (cf. Fig. 3).

Multiplicativity of Lebesgue measures together with our normalization yields

$$\frac{\mathrm{Vol}(P \times Q)}{(i + j)!} = \frac{\mathrm{Vol}(P)}{i!} \cdot \frac{\mathrm{Vol}(Q)}{j!} \,.$$

$\square$

**Lemma 6.** *Let $P, Q \subset \mathbb{R}^n$ be two lattice polytopes with $i = \dim(P)$ and $j = \dim(Q)$. If $\dim(\mathrm{conv}(P \cup Q)) = \dim(P) + \dim(Q) + 1$ then*

$$\mathrm{Vol}(P) \cdot \mathrm{Vol}(Q) \ \ divides \ \ \mathrm{Vol}(\mathrm{conv}(P \cup Q)) \,.$$

*Proof.* This is essentially the same proof, just replace the product $P \times Q$ by the join

$$P \star Q := \mathrm{conv}\left(P \times \{0\} \times \{0\} \ \cup \ \{0\} \times Q \times \{1\}\right) \ \subset \ \mathbb{R}^n \times \mathbb{R}^n \times \mathbb{R} \,.$$

(Cf. Fig. 4.)  $\square$

**Theorem 8.** *A positively homogeneous CPWL function $f$ can be represented by an integral $k$-hidden-layer neural network if and only if $f = g - h$ for two convex, positively homogeneous CPWL functions $g$ and $h$ with $P_g, P_h \in \mathcal{P}_k$.*

*Proof.* We use induction on $k$. The statement is clear for $k = 0$ because both properties apply precisely to linear functions with integral coefficients, settling the induction start.

For the induction step, assume that the equivalence is true up to $k - 1$ hidden layers for some $k \geq 1$. We show that it is true for $k$ hidden layers, too.

First, focus on the forward direction, that is, suppose $f$ can be represented by an integral $k$-hidden-layer neural network. Using Zhang et al. (2018, Lemma 6.2), $f$ is the difference (tropical quotient) of two tropical polynomials $g$ and $h$. Moreover, by the same lemma, $P_g$ and $P_h$ arise as weighted Minkowski sums of convex hulls of pairs of Newton polytopes associated with $(k - 1)$-hidden-layer neural networks. By the induction hypothesis, these polytopes are contained in $\mathcal{P}_{k-1}$. By (1), this implies that $P_g$ and $P_h$ are contained in $\mathcal{P}_k$, where the fact that the weights of the neural network are integers ensures that the resulting polytopes are lattice polytopes. This completes the first direction.

For the converse, suppose that $f$ can be written as $g - h$ with $P_g, P_h \in \mathcal{P}_k$. Using Hertrich (2022, Prop. 3.34) and (1), this implies that $g$ and $h$ can be written as sums of maxima of pairs of convex CPWL functions with Newton polytopes in $\mathcal{P}_{k-1}$. By induction, these functions can be expressed with $(k - 1)$-hidden-layer neural networks. The additional maxima and sum operations to obtain $g$ and $h$ can be realized with a single additional hidden layer (compare Hertrich (2022, Prop. 2.2)). The fact that $\mathcal{P}_k$ contains only lattice polytopes ensures that the resulting neural network has only integral weights. Since $g$ and $h$ can be expressed with $k$ hidden layers, the same holds true for $f$.  $\square$

**Lemma 11.** *One can choose $\alpha$ and $\beta$ such that $\mathrm{Lin}\, F_c \cap \mathrm{Lin}\, G_c = \{\mathbf{0}\}$ and $\mathrm{Aff}\, F_c^{\mathbf{0}} \cap \mathrm{Aff}\, G_c^{\alpha, \beta} = \emptyset$ for every $c \in \mathbb{R}^n \setminus \{0\}$.*

*Proof.* We will show that the set of parameters $\alpha$ and $\beta$ for which these conditions are not satisfied is a measure-zero set.

Let us first focus on the first statement. Suppose there is a direction $c \in \mathbb{R}^n \setminus \{0\}$ such that $\operatorname{Lin} F_c \cap \operatorname{Lin} G_c \supsetneq \{\mathbf{0}\}$ is a subspace of dimension at least one. By the definitions of $F_c$ and $G_c$ we obtain $c \in (\operatorname{Lin} F_c)^\perp$ as well as $c + \alpha \in (\operatorname{Lin} G_c)^\perp$. This implies $\alpha \in (\operatorname{Lin} F_c)^\perp + (\operatorname{Lin} G_c)^\perp = (\operatorname{Lin} F_c \cap \operatorname{Lin} G_c)^\perp$, which is a subspace of dimension at most $n - 1$, and thus a measure-zero set. Hence, choosing $\alpha$ such that it is not contained in $(\operatorname{Lin} F \cap \operatorname{Lin} G)^\perp$ for all pairs of faces $F$ and $G$ with $\operatorname{Lin} F \cap \operatorname{Lin} G \supsetneq \{\mathbf{0}\}$ guarantees that the first condition holds.

For the second statement, we choose $\alpha$ satisfying the first statement and observe that for all $c \in \mathbb{R}^n \setminus \{0\}$ we have $\operatorname{Lin} F_c \cap \operatorname{Lin} G_c = \{\mathbf{0}\}$. This implies $|\operatorname{Aff} F_c \cap \operatorname{Aff} G_c| \leq 1$ and hence $\left| \operatorname{Aff} F_c^{\mathbf{0}} \cap \operatorname{Aff} G_c^{\alpha,\beta} \right| \leq 1$. Suppose there is a direction $c \in \mathbb{R}^n \setminus \{0\}$ for which $\operatorname{Aff} F_c^{\mathbf{0}} \cap \operatorname{Aff} G_c^{\alpha,\beta} = \{(x,y)\} \subseteq \mathbb{R}^n \times \mathbb{R}$. By the choice of our lifting function from $\mathbb{R}^n$ to $\mathbb{R}^{n+1}$, it follows that $0 = y = \alpha^\top x + \beta$. Hence, choosing $\beta$ such that $0 \neq \alpha x + \beta$ for all pairs of lower faces $F^{\mathbf{0}}$ of $P^{\mathbf{0}}$ and $G^{\alpha,\beta}$ of $Q^{\alpha,\beta}$ with $\operatorname{Aff} F^{\mathbf{0}} \cap \operatorname{Aff} G^{\alpha,\beta} = \{(x,y)\}$ guarantees that the second condition holds. $\qquad\square$

**Proposition 9.** *For two polytopes $P$ and $Q$ in $\mathbb{R}^n$, there exists a subdivision of $P + Q$ such that each full-dimensional cell is an affine product of a face of $P$ and a face of $Q$.*

*Proof.* Without loss of generality, we can assume that $P + Q$ is full-dimensional. Choose $\alpha$ and $\beta$ according to Lemma 11 and consider the subdivision induced by projecting down the lower faces of $P^{\mathbf{0}} + Q^{\alpha,\beta}$. Let $C$ be a cell of dimension $n$ in the subdivision originating as a projection of a lower face $\tilde{C}$ of $P^{\mathbf{0}} + Q^{\alpha,\beta}$, minimizing a direction $(c^\top, 1) \in \mathbb{R}^n \times \mathbb{R}$. It follows from basic properties of the Minkowski sum that $\tilde{C} = F_c^{\mathbf{0}} + G_c^{\alpha,\beta}$ and thus $C = F_c + G_c$. By Lemma 11 it follows that $n \geq \dim F_c + \dim G_c \geq \dim C = n$. Hence, $C$ is an affine product of $F_c$ and $G_c$. $\qquad\square$

**Proposition 10.** *For two polytopes $P$ and $Q$ in $\mathbb{R}^n$, there exists a subdivision of $\operatorname{conv}\{P \cup Q\}$ such that each full-dimensional cell is a join of a face of $P$ and a face of $Q$.*

*Proof.* Without loss of generality, we can assume that $\operatorname{conv}\{P \cup Q\}$ is full-dimensional. Choose $\alpha$ and $\beta$ according to Lemma 11 and consider the subdivision induced by projecting down the lower faces of $\operatorname{conv}\{P^{\mathbf{0}} \cup Q^{\alpha,\beta}\}$. Let $C$ be a cell of dimension $n$ in the subdivision originating as a projection of a lower face $\tilde{C}$ of $\operatorname{conv}\{P^{\mathbf{0}} \cup Q^{\alpha,\beta}\}$, minimizing a direction $(c^\top, 1) \in \mathbb{R}^n \times \mathbb{R}$. If $\tilde{C}$ contains only vertices from either $P^{\mathbf{0}}$ or $Q^{\alpha,\beta}$, then $C$ is a face of either $P$ or $Q$ and we are done. Otherwise, it follows that $\tilde{C} = \operatorname{conv}\{F_c^{\mathbf{0}} \cup G_c^{\alpha,\beta}\}$ and thus $C = \operatorname{conv}\{F_c \cup G_c\}$. By Lemma 11 it follows that $n = \dim C = \dim \tilde{C} = \dim F_c^{\mathbf{0}} + \dim G_c^{\alpha,\beta} + 1 = \dim F_c + \dim G_c + 1$. Hence, $C$ is a join of $F_c$ and $G_c$. $\qquad\square$

**Proposition 14.** *For $P, Q \in \mathcal{Q}_k$, we have that $\operatorname{conv}(P \cup Q) \in \mathcal{Q}_{k+1}$.*

**Lemma 15.** *If $d := \dim(\operatorname{conv}(P \cup Q)) \geq 2^{k+1}$, then $\operatorname{Vol}(\operatorname{conv}(P \cup Q))$ is even.*

*Proof of Lemma 15.* By Proposition 10 it follows that $\operatorname{conv}\{P \cup Q\}$ can be subdivided such that each full-dimensional cell $C$ is a join of a face $F$ of $P$ and a face $G$ of $Q$. Let $i := \dim(F)$ and $j := \dim(G)$, then $d = i + j + 1$. By Lemma 6, we get that $\operatorname{Vol}(C) = z \cdot \operatorname{Vol}(F) \cdot \operatorname{Vol}(G)$ for some $z \in \mathbb{Z}$. Since $d$ is at least $2^{k+1}$, either $i$ or $j$ is at least $2^k$. Therefore, either $\operatorname{Vol}(F)$ or $\operatorname{Vol}(G)$ must be even because $P$ and $Q$ are contained in $\mathcal{Q}_k$. Therefore also $\operatorname{Vol}(C)$ is even. Doing so for all cells in the subdivision implies that $\operatorname{Vol}(\operatorname{conv}(P \cup Q))$ is even. $\qquad\square$

*Proof of Proposition 14.* Again, to prove the proposition, we need to ensure that not only the convex hull $\operatorname{conv}(P \cup Q)$ has even normalized volume, but each of its faces of dimension at least $2^{k+1}$. If $F$ is such a face, let $H$ be a hyperplane containing $F$. Observe that $F$ is the convex hull of all vertices of either $P$ or $Q$ that lie within $H$. In particular, $F = \operatorname{conv}(P' \cup Q')$, where $P'$ and $Q'$ are the (possibly empty) faces of $P$ and $Q$ defined via the hyperplane $H$, respectively. Since $\mathcal{Q}_k$ is closed under taking faces, it follows that $P', Q' \in \mathcal{Q}_k$. Hence, applying Lemma 15 to $P'$ and $Q'$ yields that $F$ has even volume. Doing so for all faces $F$ with dimension at least $2^k$ completes the proof. $\qquad\square$

