# OpenReview forum: "Lower Bounds on the Depth of Integral ReLU Neural Networks via Lattice Polytopes"
_ICLR.cc/2023/Conference — ICLR 2023 poster_

### Official Review · Reviewer_U42b · 2022-10-23

**Confidence:** 2
**Correctness:** 4
**Technical Novelty And Significance:** 4
**Empirical Novelty And Significance:** Not applicable
**Recommendation:** 8

**Clarity, Quality, Novelty And Reproducibility:**

- The paper is clearly written, though not completely self-contained (not possible given the topic though)
- The paper is of very high quality and novel to my knowledge

**Strength And Weaknesses:**

Strengths.
- the paper gives a thorough and long introduction into the preliminaries (until mid of p.7); this is great to give a more self-contained presentation of the topic.
- the paper proposes a novel theoretical insight into neural networks
- limitations of the study are clearly stated

Weaknesses - only minor things:
- as the authors state clearly in the paper the analysis is limited to the case of integer-valued neural networks. It is unclear how to extend the analysis to the general case, but the methods provided might be instructive towards this target.
- p.4. in conv(X): it should probably read 'lambda_i \geq 0 \forall i'?
- the structure of the paper doesn't conform to typical ICLR papers, in particular, a Conclusion section is missing.

**Summary Of The Paper:**

The paper "Lower Bounds on the Depth of Integral ReLU Neural Networks via Lattice Polytopes" derives a theoretical result and proves that log_2(n) hidden layers are necessary to compute the maximum of n input numbers. This is achieved by building on previous theoretical work that links neural networks to tropical geometry, and uses this connection to link neural networks with integer weights to lattice polytopes. Employing subdivisions of these polytopes, Minkowski sums and convex hulls, allows to construct all polytopes corresponding to a neural network of a given depth. Further, by looking at the normalized volume, it shown that not all lattice polytopes can be constructed that way, which implies a lower-bound on the depth of neural network (log_2(n)) to calculate the maximum of n numbers - matching the previously known upper bound.

**Summary Of The Review:**

The paper provides a novel theoretical insight into neural networks, and a proof construction that is well laid out.

---

> ### Author Response · Authors · 2022-11-09
> **Follow-up question about review**
>
> Thank you for taking the time to read our paper and providing valuable feedback. We are in the process of creating a revision and writing a detailed response to all issues raised by all reviewers. In the meantime, let us ask a follow-up question about your review. We would appreciate if you could reply as soon as possible such that we can incorporate the answer into our revision.
>
> We understand that most ICLR Papers contain a Conclusions section, which is probably crucial for papers with an empirical component. Our feeling is that for a purely theoretical paper like ours, where we discuss all results, their significance, limitations, and open questions already in the introduction, it is redundant to incorporate an extra section for conclusions. We are unsure about the additional value of adding such a section. Could you please comment further on that? If you feel it adds significant extra value, we would be happy to do add it to the paper. Otherwise we would prefer to stick to our style with all discussion unified in the introduction.

---

> > ### Comment · Reviewer_U42b · 2022-11-15
> > **Reply to Follow-Up Question**
> >
> > In my experience the conclusion section usually is a bit redundant with the rest of the paper, often summarizing many findings.
> > I believe it helps readability if a paper fits the typical schema - but to me it's not necessary to fit the form if the content is good.

---

> ### Author Response · Authors · 2022-11-18
> **Response to Review**
>
> We thank the reviewer for the invested time to assess our paper and provide valuable feedback and for the response to our follow-up question. We are pleased that the reviewer supports acceptance of our paper. We addressed the issue about the definition of the convex hull. Based on your response to our follow-up question, we decided to maintain our style and not add a conclusion section. Please have a look at our general comment summarizing all changes we made in our revision to implement the suggestions of all reviewers.

---

### Official Review · Reviewer_iGLm · 2022-10-24

**Confidence:** 4
**Correctness:** 3
**Technical Novelty And Significance:** 3
**Empirical Novelty And Significance:** Not applicable
**Recommendation:** 6

**Clarity, Quality, Novelty And Reproducibility:**

The manuscript’s proofs are well-written and complete. The authors described most of the manuscript with a good quality and clarity. Furthermore, I found the general concept of finding a logarithmic lower bound for the ReLU networks interesting and of value.

**Strength And Weaknesses:**

This general topic is appropriate for the machine learning community and the manuscript’s proofs are well-written. I found the general concept of finding a logarithmic lower bound for the ReLU networks interesting and of value, since finding a lower bound for this seems to be the first and the ReLU networks are the most frequently used activation function in the world right now. While I did find this work interesting, there are several concerns to be addressed before further consideration.

1-	Regarding Theorem 3’s proof, I could not find any precise justification for the used reasoning. It will be better to propose a more systematic procedure for the proof of this theorem and give a more detailed explanation of how “The arguments in Hertrich et al. (2021) can be adapted to show that the equivalence between the two conjectures is also valid in the integer case”

2-	“Translating this result back to the world of CPWL functions computed by neural networks, we obtain that k hidden layers are not sufficent” clause used in the manuscript needs more discussion about how they assumed it. Also, it has a typo. (Sufficient)

3-	The “FURTHER RELATED WORK” section is just naming some papers without a brief description. It would be better if for some cases the author gives a brief description in manuscript.

4-	It would be better if the author adds the proof of Theorem 8 to the appendix, since the paper should be self completed.

5-	The theorem 12 proof is better to be more accurate and complete.


**Summary Of The Paper:**

This paper aims to provide a non-constant lower bound on the depth of ReLU neural networks without any restriction on the width. Mostly, the authors prove that the conjecture by Hertrich et al. (2021) is true for all n ∈ N with an additional assumption on the weighs of the network to be integer numbers.

**Summary Of The Review:**

Although I found the general concept of finding a logarithmic lower bound for the ReLU networks interesting and of value, there are several concerns needs to be addressed before publication.

---

> ### Author Response · Authors · 2022-11-09
> **Follow-up questions about review**
>
> Thank you for taking the time to read our paper and providing valuable feedback. We are in the process of creating a revision and writing a detailed response to all raised issues. Before we do so, we have some follow-up questions about your review. We would appreciate if you could reply to those as soon as possible such that we can incorporate the answers into our revision.
>
> Regarding your point 1: Could you please elaborate a bit more on this aspect? Theorem 3 is the main theorem in our paper and we spend a lot of effort to prove it. While the actual proof is on page 8 and appears to be short, it is the point where we combine everything else we prove in this paper. We also have a full subsection explaining the overall proof strategy, namely Subsection 1.2. The sentence “The arguments in Hertrich et al. (2021) can be adapted to show that the equivalence between the two conjectures is also valid in the integer case” does not refer to the proof of Theorem 3, it refers to the fact that Theorem 3 implies Corollary 4. We also provide a formal proof of Corollary 4 based on Theorem 3 on the bottom of page 8.
> Given this information, could you please clarify what we should do in order to address your point 1?
>
> Regarding your point 5: Could you please also clarify what is your precise criticism and suggestion for improvement here? From our perspective, the proof of Theorem 12 based on the Propositions 13 and 15 is just a standard mathematical induction, which we clearly state.

---

> ### Author Response · Authors · 2022-11-18
> **Response to Review**
>
> We thank the reviewer for the invested time to assess our paper and provide valuable feedback. We are pleased that the reviewer tends to support acceptance. We created a revision in which we gave our best to address the concerns raised by the reviewer. Please have a look at our general comment summarizing all changes we made in our revision to implement the suggestions of all reviewers.
>
> Replies to the specific concerns:
>
> 1 - Theorem 3 is our main result and its proof is the core of our paper. We reformulated the presentation in Section 1.1 to make this clearer. Already our initial submission contained a proof overview in Section 1.2 (which we also improved in the revision). The final proof, where we combine everything, is in Section 4. The sentence “The arguments in Hertrich et al. (2021) can be adapted to show that the equivalence between the two conjectures is also valid in the integer case” does not refer to the proof of Theorem 3, it refers to the fact that Theorem 3 implies Corollary 4. We also provide a formal proof of Corollary 4 based on Theorem 3 in Section 4 (already in the initial submission). We hope that our changes are sufficient to resolve the reviewer's concerns on this point.
>
> 2 - We clarified that this is due to the duality between CPWL functions and polytopes, and added a reference to Thm. 8 where this is formally stated. We think this should be sufficient at this point since the purpose of the section is only to give a high-level proof overview.
>
> 3 - We reorganized the literature section such that papers are attributed to concrete statements. We know that especially the results about depth-width tradeoffs still look like a huge collection, but we think a detailed overview of this exciting stream of research is beyond the scope of our paper.
>
> 4 - Done.
>
> 5 - Here we really do not know what you mean. From our perspective, the proof of Theorem 12 (now 16) based on the Propositions 13 and 15 (now 12 and 14) is just a standard mathematical induction, which we clearly state. Unfortunately, we did not receive a response to our follow-up question, so we left the respective proof unchanged in our revision.
>
> We hope that our revision could resolve your concerns and we would appreciate if you would consider raising your score to support acceptance of the paper.

---

### Official Review · Reviewer_TovE · 2022-10-25

**Confidence:** 3
**Correctness:** 4
**Technical Novelty And Significance:** 3
**Empirical Novelty And Significance:** Not applicable
**Recommendation:** 6

**Clarity, Quality, Novelty And Reproducibility:**

The writing is clear and the problem is important but some of the techniques may be tied to the specific assumption of integral ReLU networks.

**Strength And Weaknesses:**

Strengths:

 It is an open problem to characterize the necessity of $\lceil \log(n+1) \rceil$-depth for dimensions $n \geq 4$ and also to get depth separations in high-dimensions, that are not constant depth, such as depth 3 vs depth 2 as shown in many works such as (Eldan and Shamir 2016, Daniely 2017, Safran et al.2019). This work tries to address this problem and gives a partial resolution and increased evidence for this. It does this by providing connections to the high-dimensional convex geometry of lattices.

Weakness:
The main weakness is that of the integrality assumption. Now as the author points out that this is impractical in training and that perhaps new techniques are needed to deal with real-valued weights. The main issue is that the separation is argued by saying that a ReLU network of a depth smaller than this threshold does not exactly represent the hard function that is provided. However, results such as (Telgarsky, 2016) focus on L1-inapproximability by networks whose depths are $o(threshold-depth)$ and that have polynomial width. Could the authors clarify if allowed for real-valued weights, if they could still get separation on depths that are say $o(\log(n+1))$ and show a stronger inapproximability result, rather than the exact representation of the function? Could the authors comment if ReLU(k), where $k=\log(n)$ and allowed for real-valued weights can still provide an arbitrarily good approximation of the hard function?

**Summary Of The Paper:**

In (Arora et al. 2018), it was shown that $\lceil \log(n+1) \rceil$-depth was sufficient for representing $n-dim$ CPWL functions. Is this necessary? Authors say yes, under the assumptions that weights are integral. That is, there exists functions that cannot be represented by depth less than $\log(n+1)$, under the conditions that the weights are integral. To show this they exploit the fact that the weights are integral and the hard function is $\text{max}[0,x_1,x_2,\ldots,x_n]$, where $n=2^k$ and this is not expressible by a ReLU with integral weights and depth $k \leq \log(n)$.

**Summary Of The Review:**

Overall, I think it is a good attempt at a big open question w.r.t the depth separation results, but I feel there are some questions as mentioned above, that requires some explanation and probably a bit more work.

---

> ### Author Response · Authors · 2022-11-18
> **Response to Review**
>
> We thank the reviewer for the invested time to assess our paper and provide valuable feedback. We added the references mentioned by the reviewer to the paper in case they were not already there. Please have a look at our general comment summarizing all changes we made in our revision to implement the suggestions of all reviewers.
>
> While we do not address questions about inapproximability and exponential benefits of depth over width, as correctly pointed out in the report, we focus on the exact representability with arbitrary width. We hope that our additional elaborations in our revision help to clarify this slightly different point of view.
> In the following, we give more detailed answers to the concerns raised in the report explaining the importance of our perspective.
>
> We agree that the main weakness is the integrality assumption. However, as we argue in our Section 1.3 (Beyond Integral Weights), while there are definitely new ideas needed to address the conjecture by Hertrich et al. (2021) in full generality, we are also convinced that the techniques in this paper do indeed pave the way towards this goal.
>
> Furthermore, we agree that our paper does not make any claim about (and has no direct implications for) inapproximability. In fact, the question we ask has a simple answer in an approximate setting: we allow arbitrary finite width (and do not restrict ourselves to polynomial width, as mentioned in the review), in which case universal approximation theorems already guarantee approximability for shallow neural networks. Nevertheless, we think that exploring such a fundamental question as "what are actually the precise functions represented by neural networks with a certain depth" is important to gain a better mathematical understanding of deep learning. To give an example beyond that, understanding the exact piecewise linear structure of neural networks has proven to be very helpful in determining the computational complexity of neural network training, see e.g., Arora et al. (ICLR 2018), Bertschinger et al. (2022), and other references we give in the paper. Moreover, we concieve that, on the long run, such questions will form synergies with the approximation point of view.
>
> Specific answers to the questions of the reviewer:
> - No, we do not obtain such a separation. We hope that we could clarify that this was not one of our goals.
> - Yes, universal approximation theorems imply that such an approximation is possible, at least on a compact domain.
>
> We hope that our response and our revision could sufficiently address your concerns and we would appreciate if you would consider raising your score in order to support acceptance of the paper.

---

> > ### Comment · Reviewer_TovE · 2022-11-24
> > **Thank you for the clarifications.**
> >
> > Thank you for the responses, yes I believe the changes you made clarify and position this result in the plethora of the "depth-separation" results for neural nets and thus I feel that this is a good work w.r.t integral ReLU networks. However, focusing on exact representations and integral weights slightly limits the scope in my view. I will increase my score to 6 in light of this.

---

### Official Review · Reviewer_dCJW · 2022-10-29

**Confidence:** 4
**Correctness:** 3
**Technical Novelty And Significance:** 3
**Empirical Novelty And Significance:** Not applicable
**Recommendation:** 8

**Clarity, Quality, Novelty And Reproducibility:**

The paper is mostly clear, writing can be improved though (see below). The results are novel, and the proofs seem correct.

Suggestions for improving writing:
- All figures: Add descriptions in the caption to explain the figures, they are not easy for a reader without background to understand
- Better notation for $\tilde{L}$ parallel to $L$ (just above section 2.3)
- Proof of Theorem 3 uses Proposition 17 before it is even defined. Please reorganize to ensure a linear flow. Also add more explanations of the proof, and high level ideas.

Questions:
- How does the integer weight assumption compare to the condition used in Hertrich et al. 2021? Are they comparable?
- Have you explored the width and depth tradeoff for these results?

**Strength And Weaknesses:**

**Strengths**:
- The paper takes a good step towards understanding the depth separation of exact function computation for ReLU networks. Unlike approximation results where one non-linear layer suffices for approximating functions, the question of exact representation is not so straightforward. Though the paper does not solve the question in its generality, understanding even the integer weight constraint is useful, and hopefully helpful for the general setting.
- Showing separation for max of linear functions is insightful since it suggests that adding max-pooling layers (which are popular in practice) might be a computationally useful operation.

**Weaknesses**:
- Despite being mathematically interesting, exact representability is not the most interesting from the point of learning in-distribution. Approximate approximation suffices for that.
- There is no mention of width of the network. As has been seen in Telgarsky 2015, width and depth do have their interesting trade-offs.
- The overall writing of the paper is not very friendly to readers without a tropical geometry background.

**Summary Of The Paper:**

The paper studies the depth requirements for exactly representing the maximum of $n$ inputs using networks with integer weights. They show that $\lceil \log_2 n \rceil$ depth is necessary. Note that the simple binary tree construction that computes maximum of two inputs at a time achieves this depth. For this class of integer weight NNs this shows a depth separation result for all depths less than $\lceil \log_2 n \rceil$. The lower bound sheds light on the importance of using max pooling layers to remove depth dependence.

**Summary Of The Review:**

Overall, I found the problem studied in the paper and the results mathematically interesting. I think they are a nice addition to the rich literature of representation results known for neural networks despite the limitation of exact representability. Therefore, I vote to accept the paper. I encourage the authors to improve the writing.

---

> ### Author Response · Authors · 2022-11-18
> **Response to Review**
>
> We thank the reviewer for the invested time to assess our paper and provide valuable feedback. We are pleased that the reviewer supports acceptance of our paper. Please have a look at our general comment summarizing all changes we made in our revision to implement the suggestions of all reviewers.
>
> Let us quickly comment on the weaknesses mentioned by the reviewer:
>
> - We agree that our results have no direct implication on approximation abilities of neural networks. Nevertheless, we think that exploring such a fundamental question as "what are actually the precise functions represented by neural networks with a certain depth" is important to gain a better mathematical understanding of deep learning. To give an example beyond that, understanding the exact piecewise linear structure of neural networks has proven to be very helpful in determining the computational complexity of neural network training, see e.g., Arora et al. (ICLR 2018), Bertschinger et al. (2022), and other references we give in the paper. Moreover, we concieve that, on the long run, such questions will form synergies with the approximation point of view.
>
> - Concerning your second mentioned weakness, please refer to our answer to the second question below.
>
> - We hope our revision addresses the main concerns raised by the reviewer with respect to writing.
>
> Now let us answer the specific questions by the reviewer:
>
> - Even though our integrality assumption and the assumption by Hertrich, Basu, Di Summ, Skutella (2021), called H-conforming, originate from similar intuitions, they are incomparable: either one can be satisfied while the other one is violated. We added a short discussion of this at the beginning of the literature section.
>
> - While our result does not give a depth-width tradeoff in the classical sense (something like "depth is exponentially more important than width"), our result does indeed imply a very fundamental and related tradeoff, separating realizibility and non-realizibility with any finite width from each other: while finite width is always sufficient with logarithmic depth, there are functions which can never be represented with sublogarithmic depth and any finite width. We added a short discussion of this at the respective point in the literature section.

---

> > ### Comment · Reviewer_dCJW · 2022-11-21
> > **Response to Rebuttal**
> >
> > Thank you for addressing my concerns, and improving the presentation of the paper. I will maintain my score, and continue to vote for the paper to be accepted.

---

### Author Response · Authors · 2022-11-18
**Key Changes in Revision**

We thank all four reviewers for their invested time to assess our paper and provide valuable feedback. We reply to the points raised in the reviews separately below each review. Based on the reviews we created a revision. The key changes in the revision are:

- Section 1.1 (Our Results): We improved the presentation, making it clearer that Theorem 3 is our main result and its proof is the core of our paper. We added references to Section 1.2 (proof overview) and Section 4 (the actual proof).
- Section 1.2 (Outline of the Argument): We improved the presentation, adding references to various sections and statements in the paper, and sharpening some formulations to address concerns by Reviewer iGLm.
- Literature section: We added a comparison with the assumption (H-conforming) used by Hertrich, Basu, Di Summa, Skutella (NeurIPS 2021).
- Literature section: We rewrote the section to make it clearer which contribution was made by which paper and added the suggested references by reviewer TovE.
- We improved all figure captions, making them more accessible.
- We now denote affine lattices by $K$ instead of $\tilde{L}$.
- We restructured Section 4 to ensure a "linear flow", that is, we do not use statements before defining them any more.
- At the beginning of Section 4 we added an additional overview of the proof strategy and a high-level intuition behind most of the proofs of the section.
- We added a proof of Thm. 8 to the appendix for the sake of self-containment.
- We moved the proofs of Prop. 14 and Lem. 15 (new numbering) to the appendix to make space for all the other changes.

---

### Decision · Program_Chairs · 2023-01-20

**Decision:**

Accept: poster

**Justification For Why Not Higher Score:**

The assumptions can be quite restrictive (i.e., the integrality) though they provide insights.

**Justification For Why Not Lower Score:**

The work is a solid theoretical attempt on understanding the expressivity of NNs.

**Metareview: Summary, Strengths And Weaknesses:**

This is a theory paper on the expressivity of neural networks using lattice polytope perspective. The authors prove that log2(n) hidden layers are necessary to compute the maximum of n input numbers.

The paper is well-written and manages to do a great job in introducing the preliminaries while being self-contained.

The integrality assumption is the main weakness of the work.

**Note From Pc:**

if the above contains the word "oral" or "spotlight" please see: "oral" presentation means -> notable-top-5% and "spotlight" means -> notable-top-25%. As stated in our emails, we are disassociating presentation type from AC recommendations